# Variance Matters: Detecting Semantic Differences without Corpus/Word Alignment

**Ryo Nagata**
Konan University, Japan
nagata-emnlp2023 @ml.hyogo-u.ac.jp.

**Hiroya Takamura**
AIST, Japan
takamura.hiroya@aist.go.jp

**Naoki Otani**
Tokyo University of Foreign Studies, Japan
otani@tufs.ac.jp

**Yoshifumi Kawasaki**
University of Tokyo, Japan
ykawasaki@g.ecc.u-tokyo.ac.jp

## Abstract

In this paper, we propose methods[1] for discovering semantic differences in words appearing in two corpora. The key idea is to measure the coverage of meanings of a word in a corpus through the norm of its mean word vector, which is equivalent to examining a kind of variance of the word vector distribution. The proposed methods do not require alignments between words and/or corpora for comparison that previous methods do. All they require are to compute variance (or norms of mean word vectors) for each word type. Nevertheless, they rival the best-performing system in the SemEval-2020 Task 1. In addition, they are (i) robust for the skew in corpus sizes; (ii) capable of detecting semantic differences in infrequent words; and (iii) effective in pinpointing word instances that have a meaning missing in one of the two corpora under comparison. We show these advantages for historical corpora and also for native/non-native English corpora.

## 1 Introduction

In this paper, we propose a method for detecting semantic or usage differences[2] in words and also a method for extracting their typical instances in context based on variance of contextualized word vectors, which bring out various applications: e.g., historical linguistics (Hamilton et al., 2016) (e.g., discovering words that have acquired a meaning) and second language acquisition research (McEnery et al., 2019) (e.g., words and their meanings that non-native speakers do not use as native speakers). The key observation is that the more meanings a word covers in a corpus, the shorter its mean word

vector becomes. We show that this property of contextualized word vectors can be quantified as a kind of variance of their distribution and that it is a good indicator of semantic differences.

The major approach to semantic difference detection, which is based on non-contextualized word vectors such as Word2vec (Mikolov et al., 2013), has several limitations and thus is not always applicable to any corpora as will be discussed in detail in Sect. 5. Above all, many non-contextualized word vector-based methods require some sort of correspondence between two corpora for comparison (e.g., word alignment). The task is, however, to find words that do not correspond well in terms of their meanings, and thus it is more natural not to assume any correspondence in advance, as Aida et al. (2021) point out. For example, it is not straightforward at all to align words between native and non-native English corpora. Besides, most previous methods are computationally costly and are not suitable for targeting all words in a large corpus.

In contrast, the proposed methods do not assume any correspondence between corpora. All they require are to compute the mean of contextualized word vectors and its variance (or its norm) for each word type. They do not require training or hyper-parameter search unlike previous methods. Nevertheless, they rival the best-performing system in the SemEval-2020 Task 1 (Schlechtweg et al., 2020). They are even effective in corpus pairs whose sizes are considerably different and also in infrequent words. In addition, they are capable of pinpointing word instances that have a meaning missing in one of the two corpora under comparison. For instance, in Sect. 3, they reveal that *near* is one of the most semantically different words between the native and non-native sub-corpora (approximately 10,000 and 100,000 words, respectively) of ICNALE (Ishikawa, 2011); its most representative instance out of the 11 *near* occurrences in the native portion is "*it has near impossible*," which is

---

[1]The source codes are available at https://github.com/nagata-github/vmf_meaning_change_detector/

[2]Following the convention in the literature, we use the term *semantic difference* rather abstractly to refer to differences in meaning and usages. In this paper, semantic meanings of linguistic units include not only lexical meanings, but also contextual meanings, discourse functions, and grammatical functions.

interpreted as *almost*; this usage does not appear at all in the 267 instances of *near* in the non-native sub-corpus.

The contributions of this paper are three-fold: (i) We show for the first time that norms of the mean contextualized word vectors are a good indicator of semantic differences; (ii) We show that (i) is mathematically equivalent to examining variance of the word vector distribution; (iii) We actually reveal words that have semantic differences with their representative instances in 1800s/2000s English and also in native/non-native English.

## 2 Methods

We describe two methods, one for detecting words that have semantic differences in two corpora and one for extracting their representative instances. So far, we have used the term *word* abstractly to mean both *word type* and *word token*. Hereafter, for better understanding, we will distinguish between the two; we will use the term *word type* to refer to word types and the term *word instance* to refer to word tokens in context, which we assume is a whole sentence.

### 2.1 Detecting Semantic Differences

To begin with, let us first note that the similarity between two words (tokens or types) are conventionally measured by the cosine similarity between their word vectors (hereafter, for simplicity, word vectors will refer to contextualized ones unless otherwise noted). This is equivalent to measuring the word similarity based only on the directions of word vectors, or to assuming that all word vectors are normalized so that their norms become one. We follow this convention, hereafter.

Under this condition, any word vector appears on the unit hypersphere. As a special case of this, when the dimension of word vectors is two, word vectors appear on the unit circle as in the dashed arrows (vectors) in Fig. 1.

We now examine the norm of the mean word vector for various cases. An extreme case would be that a word type is always used in the exact same context, and thus with the same meaning. Its word vectors appear at the same point on the unit hypersphere as in Fig. 1 (a). Then, its mean vector is always identical to the original word vectors, and thus its norm is also always one; recall all word vectors are normalized so that their norms equal one. The other extreme case would be that a

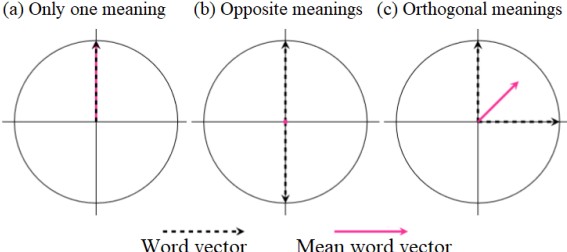

Figure 1: Intuitive Illustration for Mean Norms.

word type is represented by two opposite vectors as in Fig. 1 (b), which should cover much wider meanings. In this case, its mean vector becomes the zero-vector with the zero norm. Other cases in between would give a norm between zero and one. For instance, two orthogonal vectors result in the mean word vector whose norm is $\frac{\sqrt{2}}{2}$ as in Fig. 1 (c)[3].

The observations so far suggest that the wider meanings a word type covers in a given corpus, the shorter its mean word vector becomes. This property of word vectors is the basis of the proposed methods.

To discuss formally, we will introduce the following symbols. We will denote a word vector by $\boldsymbol{x}$. Recall once again that $\|\boldsymbol{x}\| = 1$ for all $\boldsymbol{x}$. We will also denote the mean vector of $\boldsymbol{x}$ and its norm by $\overline{\boldsymbol{x}}$ and $l$, respectively (i.e., $\overline{\boldsymbol{x}} \equiv \frac{1}{n} \sum_{i=1}^{n} \boldsymbol{x_i}$ and $l \equiv \|\overline{\boldsymbol{x}}\|$ where $n$ refers to the number of word instances of that word type in a given corpus). We will denote the two corpora for comparison by $S$ and $T$ (source and target[4], respectively); for example, $l_S$ refers to the norm of the mean word vector of a word type obtained from the source corpus.

With this notation, the straight forward implementation of the above idea for measuring semantic differences would be taking the ratios $l_T/l_S$ for all word types appearing in two corpora; larger values of this indicate larger semantic differences (wider and narrower meanings in the source and target corpora, respectively).

In the proposed method, we use its extended version as our score function, which we call *coverage*; we define the coverage as

$$c(S, T) = \log \frac{l_T(1 - l_S^2)}{l_S(1 - l_T^2)}, \quad (1)$$

the reason for which we will explain in Subsect. 2.3.

---

[3]Addition of two orthogonal vectors produces a vector along the diagonal line with a norm of $\sqrt{2}$, and thus the norm of the mean word vector is $\frac{\sqrt{2}}{2}$.

[4]Source and target corpora would, for example, be native and non-native English corpora.

For the time-being, let us just notice that in the coverage, the norms $l_T$ and $l_S$ are respectively weighted by $1 - l_S^2$ and $1 - l_T^2$, which are based on their counterpart. Also, note that taking the log does not affect rankings by the ratio and that, at the same time, positive and negative values indicate narrowing and broadening meanings of the word type, respectively.

The procedure for detecting word types having semantic differences is as follows:
**Input**: source and target corpora $S, T$
**Output**: a list of words sorted in order of coverage

1. Vectorize all word instances in $S$ and $T$
2. For each word type, compute its mean vectors $\overline{\boldsymbol{x}}_S$ and $\overline{\boldsymbol{x}}_T$, and then its norms $l_S$ and $l_T$
3. Sort the word types by the coverage defined by Eq. (1) in descending order
4. Output the sorted list

## 2.2 Extracting Typical Word Instances

We now turn our focus to extracting word instances having a meaning which is not, or seldom if ever, used in the target corpus. For this, we once again consider the illustrative unit circle shown in Fig. 2. Fig. 2 exemplifies two mean vectors $\overline{\boldsymbol{x}}_S$ and $\overline{\boldsymbol{x}}_T$ of a word type obtained from the source and target corpora, respectively. Intuitively, the word instances (or their word vectors) that we are looking for now are those that are distant from the mean word vector for the target corpus (to make sure that their meanings are not, or seldom, used in it) and also that are near the mean word vector for the source corpus (to make sure that their meanings are indeed used in it). The dashed arrow $\boldsymbol{x}_S$ shown in Fig. 2 would be an example of this.

Fortunately, a difference of the two mean word vectors (i.e., $\overline{\boldsymbol{x}}_S - \overline{\boldsymbol{x}}_T$) will facilitate satisfying these conditions. Fig. 2 illustrates that the word vector $\boldsymbol{x}_S$ in the source corpus is similar to $\overline{\boldsymbol{x}}_S - \overline{\boldsymbol{x}}_T$. This corresponds to taking:

$$\cos(\overline{\boldsymbol{x}}_S - \overline{\boldsymbol{x}}_T, \boldsymbol{x}) = \frac{(\overline{\boldsymbol{x}}_S - \overline{\boldsymbol{x}}_T)^\mathsf{T}\boldsymbol{x}}{\|\overline{\boldsymbol{x}}_S - \overline{\boldsymbol{x}}_T\|\|\boldsymbol{x}\|} \quad (2)$$

Noting $\|\boldsymbol{x}\| = 1$ and that $\|\overline{\boldsymbol{x}}_S - \overline{\boldsymbol{x}}_T\|$ is constant with respect to $\boldsymbol{x}$, Eq. (2) reduces to $(\overline{\boldsymbol{x}}_s - \overline{\boldsymbol{x}}_t)^\mathsf{T}\boldsymbol{x}$. As in Subsect. 2.1, we will adjust this cosine-based function by $1 - l_S^2$ and $1 - l_T^2$ to define another score function called *representativeness* as

$$r(\boldsymbol{x}, S, T) = (\frac{1}{1 - l_S^2}\overline{\boldsymbol{x}}_S - \frac{1}{l - l_T^2}\overline{\boldsymbol{x}}_T)^\mathsf{T}\boldsymbol{x}, \quad (3)$$

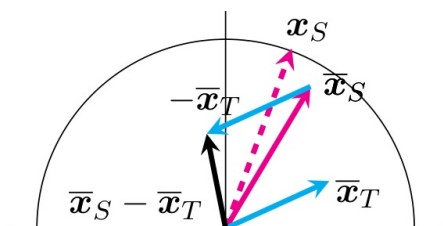

Figure 2: Illustration for Difference of Mean Vectors.

for which we will give the mathematical background in Subsect. 2.3.

The procedure for extracting word instances having a meaning which is not, or seldom used in the target corpus is as follows:
**Input**: source and target corpora $S$, $T$; a target word type $w$
**Output**: a list of word instances in $S$ sorted in order of the representativeness

1. For $w$, compute the mean word vectors $\overline{\boldsymbol{x}}_S$ and $\overline{\boldsymbol{x}}_T$ from $S$ and $T$, respectively
2. For each word instance of $w$ in $S$, compute the representativeness defined by Eq.(3)
3. Sort the word instances by the representativeness in descending order
4. Output the sorted list

The list obtained by swapping $S$ and $T$ is also helpful to investigate where the meaning difference comes from.

## 2.3 Mathematical background

We now give a mathematical background to the proposed methods. Specifically, we show that the two score functions assume the von Mises-Fisher distribution behind word vectors (see the study (Banerjee et al., 2005) for the details of the distribution).

The von Mises-Fisher distribution is a probability density function for the random $d$-dimensional unit vector $\mathbf{x}$. It is defined as

$$f(\mathbf{x}; \boldsymbol{\mu}, \kappa) = z_\kappa \exp\left(\kappa\boldsymbol{\mu}^\mathsf{T}\mathbf{x}\right). \quad (4)$$

The parameters $\boldsymbol{\mu}$ ($\|\boldsymbol{\mu}\| = 1$) and $\kappa$ ($\kappa \geq 0$) are respectively the mean direction and concentration parameter. The constant $z_\kappa$ is the normalization constant depending on $\kappa$. The distribution can be regarded as akin to the isotropic Gaussian distribution of the hypersphere. It is commonly used to process directional data as in the present paper.

In our case, the unit vector $\mathbf{x}$ of the von Mises-Fisher distribution is the word vector $\boldsymbol{x}$. It follows that the word vector $\boldsymbol{x}$ distributes isotropically

around the mean direction $\boldsymbol{\mu}$ with the concentration $\kappa$. Then, $\kappa$ is interpreted as the concentration of word meanings of the corresponding word type. Conversely, the reciprocal $1/\kappa$ can be seen as a kind of variance of word meanings.

Considering this, we define the coverage as $\log(\kappa_T/\kappa_S)$. To examine the ratio $\log(\kappa_T/\kappa_S)$, one needs to estimate $\kappa$. Banerjee et al. (2005) show a simple approximate solution of its maximum likelihood estimate is:

$$\kappa \approx \frac{l(d - l^2)}{1 - l^2}, \tag{5}$$

where $l$ is the norm of the mean vector as defined in Subsect. 2.1 while $d$ denotes the dimension of the unit vector $\mathbf{x}$. Then the ratio is approximated to

$$\log \frac{\kappa_T}{\kappa_S} \approx \log \frac{\frac{l_T(d - l_T^2)}{1 - l_T^2}}{\frac{l_S(d - l_S^2)}{1 - l_S^2}}. \tag{6}$$

Using[5] $d \gg l$, it is further approximated to

$$\log \frac{\kappa_T}{\kappa_S} \approx \log \frac{l_T(1 - l_S^2)}{l_S(1 - l_T^2)}, \tag{7}$$

which is equivalent to our score function *coverage*.

For the representativeness (Eq. (3)), we can show that it is equivalent to examining the Log Likelihood Ratio (LLR) of the probability density function, which compares how probable the given $\boldsymbol{x}$ is in the two corpora. It is given by

$$
\begin{aligned}
\mathrm{LLR} &= \log \frac{z_{\kappa_S} \exp\left(\kappa_S \boldsymbol{\mu}_S^{\mathsf{T}} \boldsymbol{x}\right)}{z_{\kappa_T} \exp\left(\kappa_T \boldsymbol{\mu}_T^{\mathsf{T}} \boldsymbol{x}\right)} \\
&= \log \frac{z_{\kappa_S}}{z_{\kappa_T}} + (\kappa_S \boldsymbol{\mu}_S - \kappa_T \boldsymbol{\mu}_T)^{\mathsf{T}} \boldsymbol{x}. \tag{8}
\end{aligned}
$$

The maximum likelihood estimate of $\boldsymbol{\mu}$ is given by $\boldsymbol{\mu} = \frac{\overline{\boldsymbol{x}}}{l}$ (Banerjee et al., 2005). Here, note that only the second term in the second line matters with respect to $\boldsymbol{x}$. Then, putting Eq. (5) into the second term results in $(\frac{d - l_S^2}{1 - l_S^2} \overline{\boldsymbol{x}}_S - \frac{d - l_T^2}{1 - l_T^2} \overline{\boldsymbol{x}}_T)^{\mathsf{T}} \boldsymbol{x}$. The approximations $d - l_T^2 \approx d - l_S^2$ for $d \gg l_T^2$ and $d \gg l_S^2$ give the score function *representativeness*. Note the coarse approximation $\kappa \approx l$ would give the naive score functions originally introduced in Subsect. 2.1 and 2.2.

---

[5]For example, $d = 1024$ when 'bert-large-uncased' is used as a vectorizer while $l \in [0, 1]$.

## 3 Evaluation

### 3.1 Data and Conditions

We detect word types having semantic differences and extract their word instances using the proposed methods to evaluate their effectiveness. Specifically, we compare the following two corpus pairs for three purposes: 1800s/2000s English for a historical analysis; native/non-native English for comparisons between native/non-native speakers of English and between non-native speakers. We use the cleaned version (Alatrash et al., 2020) of COHA (Davies, 2012) for the first one and IC-NALE (Ishikawa, 2011) for the second and third ones. Table 1 shows their sizes.

COHA provides texts published between the 1820s and the 2010s. Accordingly, we use the texts in the corresponding periods. In COHA, 5% of ten consecutive tokens every 200 are replaced by '@' due to copyright regulations. We exclude sentences containing this special token from our analysis. They also contain a wide variety of fixed labels such as citation information as in *Produced from page scans provided by Internet archive*. These inevitably make the norm of the mean word vector longer for the words. Also, they can be noise in that words would not appear in the corpora (e.g., *Internet* in the 1800s). Similarly, proper names often collocate with fixed contexts such as movie scripts (e.g., *John: Yes, it is.*). We exclude these noisy word types and proper nouns from the sorted list of word types[6].

In addition, we use the SemEval-2020 Task 1 dataset (Schlechtweg et al., 2020), which is also based on COHA, to evaluate performance in semantic difference detection quantitatively. We evaluate the proposed detection method in its English sub-task 1[7], the binary classification task of decid-

---

[6]We manually removed such words by consulting their typical word instances from the lists shown in Table 3 and Table 7 in the following sections.

[7]The official dataset provides original and lemmatized texts. The target word instances are tagged only in the latter. We semi-automatically tagged the original texts by aligning the two and used them in this experiment.

| Corpus | # tokens |
|---|---|
| COHA 1800s | 111,048,657 |
| COHA 2000s | 68,678,659 |
| ICNALE Native | 97,899 |
| ICNALE Non-native | 986,764 |

Table 1: Sizes of Corpora for Evaluation.

ing which words have a semantic difference out of the 37 target words. Since the proposed detection method is designed to predict whether meanings of a word type are narrowing or broadening, we take $\frac{\max(\kappa_T, \kappa_S)}{\min(\kappa_T, \kappa_S)}$ to perform this binary classification (the larger the value, the greater the difference, whether narrowing or broadening, a word type has); we use the $K$-means++ algorithm with $K = 2$ to determine the threshold for the binary classification.

ICNALE consists of essays written by native and non-native speakers of English. As a non-native sub-corpus, we use the essays labelled as either China, Indonesia, Japan, Korea, Taiwan, and Thailand. The essay topics are written on either (a) *It is important for college students to have a part-time job.* or (b) *Smoking should be completely banned at all the restaurants in the country.* This means that the essay topics are common to the native and non-native sub-corpora, while the former is ten times smaller than the latter, as shown in Table 1.

The other conditions in this evaluation are as follows. In all corpora, we only target tokens that occur more than ten times. We use as word vectors the outputs of the final layer of 'bert-large-cased' (Devlin et al., 2019) for the SemEval sub-task and 'bert-large-uncased' for the other tasks in the Hugging Face implementations. We only target tokens that are not split into multiple sub-words and that consist only of alphabetic letters.

### 3.2 Semantic Difference Detection

We begin by evaluating detection accuracy in the sub-task 1 of the SemEval-2020 Task 1. Table 2 shows detection accuracy of the proposed detection method together with that of the following four methods for comparison: the best-performing system (Rother et al., 2020) in the SemEval shared task; Aida et al. (2021)'s method, which is a non-alignment-based method that can be applicable to all word instances in a given corpus as the proposed method; Mean direction-based: $1 - \cos(\overline{\boldsymbol{x}}_S, \overline{\boldsymbol{x}}_T)$ is used as a score function instead of Eq.(1); Mean direction & coverage-based: the average of $1 - \cos(\overline{\boldsymbol{x}}_S, \overline{\boldsymbol{x}}_T)$ and the coverage is used as a score function.

It turns out that despite its simplicity, the proposed detection method rivals the SemEval best-performing system that involves much more complicated processes — dimensionality reduction and clustering — on top of contextualized word vectors.

| Method | Accuracy |
|---|---|
| Proposed | 0.730 |
| Rother et al. (2020) | 0.730 |
| Aida et al. (2021) | 0.676 |
| Mean direction-based | 0.622 |
| Mean direction & coverage-based | 0.702 |

Table 2: Semantic Difference Detection Accuracy in Sub-task of SemEval-2020 Task 1.

These two processes make it difficult to apply the Rother et al. (2020)'s method to all word types in a large corpus. While Aida et al. (2021)'s method requires less computation, it performs worse than the proposed detection method. Somewhat unexpectedly, the methods that exploit the mean directions do not perform as well as the proposed method depending only on the concentration parameter $\kappa$ (or equivalently variance). The results suggest that semantic differences are reflected more in variance than in the mean direction. We will discuss this point in detail in Sect. 4.

### 3.3 Comparing 2000s to 1800s English

Table 3 shows the top ten word types having wider meanings in the 2000s (source) than in the 1800s (target) corpus; for space limitation, the table for the opposite combination is shown in Appendix A. Note that "$S_i$:" and "$T_i$:" in the representative instance column denote that the corresponding word instances are respectively the $i$th word instance in the lists obtained by the representativeness from the source and target corpora.

Table 3 reveals the following two categories of word types[8]: potential semantic shift and differences in Part-Of-Speech (POS). We describe them in detail in this order below.

**Potential semantic shift**: *systemic* in Table 3 is a typical example. It is frequently used in body-related sense as in *systemic circulation* in both sub-corpora. In contrast, the representative word instance *systemic injustices* only appears in the 2000s. This usage should be a relatively new concept, probably introduced in the 21st century. A similar argument applies to a kind of technical term *political spectrum* in the 2000s, which is a system that classifies different political positions in relation to one another.

The other examples are as follows (the following interpretations in brackets are all in 2000s to 1800s

---

[8] *champs* is an exception. It is often used as a French word as in *Champs Elysees* in the 1800s while it means *champions* in the 2000s.

| $c(S,T)$ | Word type | $f_S$ | $f_T$ | Representative instance |
|---|---|---|---|---|
| 0.93 | systemic | 210 | 24 | $S_1$: *systemic* injustices / $T_1$: *systemic* circulation |
| 0.91 | dynamo | 77 | 39 | $S_3$: teen *dynamo* daughter / $T_1$: motor or reversed *dynamo* |
| 0.83 | conversions | 36 | 90 | $S_1$: *conversions* into theaters / (religious) *conversions* |
| 0.83 | trigger | 1222 | 337 | $S_2$: can *trigger* widespread problems / $T_2$: drew the *trigger* |
| 0.81 | strikers | 17 | 205 | $S_1$: Pjanic playing behind the *strikers* / $T_1$: the alacrity of the *strikers* |
| 0.78 | grille | 100 | 11 | $S_1$: bar & *grille* / $T_2$: open the *grille* |
| 0.76 | rotating | 333 | 29 | $S_1$: have been *rotating* / $T_1$: the *rotating* sphere |
| 0.73 | champs | 82 | 60 | $S_2$: national *champs* / $T_1$: *Champs* Elysees |
| 0.73 | spectrum | 618 | 272 | $S_1$: political *spectrum* / $T_5$: prismatic *spectrum* |
| 0.72 | norm | 446 | 15 | $S_1$: wider than the *norm* at Olympic / $T_1$: *norm* of correct English |

Table 3: Top Ten Word Types Having Wider Meanings in 2000s (*Source*) than 1800s (*Target*) in COHA. $f_S$ and $f_T$: word frequencies in $S$ and $T$, respectively.

order): *dynamo* (an energetic person vs. a machine), *conversions* (physical vs. religious), *strikers* (a position in football vs. people who are on strike), *grille* (an eating place/a cooking tool vs. metal bars/a gate), and *norm* (average vs. a standard model). While it is difficult to attest them from the information available, these word types, at least, are used in wider meanings in the 2000s.

**Differences in POS**: *trigger* and *rotating* fall into this category. The former is used often as a noun (a part of a gun) in the 1800s while it is used also as a verb to mean *to cause*. This can also be interpreted as a meaning shift. The latter is used mainly as an attributive adjective in the 1800s while also as a verb in the 2000s. Differences in POS like these are reflected in a large value of $c(S,T)$.

### 3.4 Comparing Native to Non-Native English

Table 4 shows the top 12 word types having wider meanings in the native sub-corpus (source) in IC-NALE, which follows the same format as Table 3; the table for the opposite combination is available in Appendix B. Table 4 reveals the following three major reasons for the semantic differences: influence from essay prompt, idiomatic phrases, and differences in construction and part-of-speech (POS).

**Influence from essay prompt**: Simply, many of the non-native speakers use one of the essay prompts *Smoking should be completely banned at all the restaurants in the country.* as it is (and its partial phrases). To be precise, the entire phrase appears 39 times and only once in the non-native and native sub-corpora, respectively. This naturally makes the contexts of *completely* and *country* rather fixed in the non-native sub-corpus, resulting in their large norms of their mean word vectors.

**Idiomatic phrases**: More interestingly, Table 4

reveals word types used in an idiomatic phrase or a phrasal verb that seldom appear in the non-native sub-corpus, including *fall into place*, *in place* (as in *effective*), and *hold down a job* (as in *manage to keep the job*). The non-native writers often use *place* to refer to physical locations while the native speakers also use it metaphorically, including the idiomatic phrases. The word *hold* appears more than 100 times in the non-native sub-corpus, but never collocates with *down*, suggesting that most non-native speakers do not use or know the phrasal verb that native speakers use (four out of the 16 instances of *hold* appear in the phrasal verb in the native sub-corpus). Instead, they often use it as a transitive verb as in *hold a job*, which also frequently appears in the native sub-corpus.

Idiomatic phrases also play the opposite role. Specifically, the non-native speakers repeatedly use *concerned* and *course* in the idiomatic phrases as shown in Table 4. Surprisingly, they use the idiom *of course* 87% of the time while the native speakers often use *course* to mean *a set of classes*, which decreases the relative frequency of the idiomatic phrase (63%). Although strictly, they are not idiomatic phrases, the non-native speakers use *first* and *third* in a fixed phrase. Surprisingly again, for instance, the first 886 word instances of *first* (sorted by $r(x,T,S)$) are actually *First, ···*. It should be emphasized that the nationalities of the non-native speakers range over six countries and nevertheless, fixed expressions like these are common to them. This is beyond the scope of this paper, but it would be interesting to investigate the reasons behind this.

**Differences in construction and POS**: These are represented by *near* and *knowledge*. The former only appears 11 times in the native sub-corpus and its representative instance is the one that can

| $c(S,T)$ | Word type | $f_S$ | $f_T$ | Representative instance |
|---|---|---|---|---|
| 0.61 | completely | 48 | 1662 | $T_1$: ESSAY PROMPT |
| 0.54 | near | 11 | 267 | $S_2$: it has become *near* impossible |
| 0.50 | country | 48 | 1707 | $T_1$: ESSAY PROMPT |
| 0.46 | concerned | 11 | 113 | $T_1$: as far as I'm *concerned* |
| 0.45 | third | 11 | 348 | $T_1$: Third, $\cdots$ |
| 0.39 | period | 13 | 115 | $S_1$: Period! |
| 0.38 | first | 87 | 1512 | $T_1$: First, $\cdots$ |
| 0.37 | course | 46 | 489 | $T_1$: Of *course* $\cdots$ |
| 0.37 | place | 67 | 1764 | $S_1$: put a ban in *place* / $S_6$: fall into *place* |
| 0.36 | taking | 34 | 461 | $S_1$: *taking* away / $T_1$: *taking* a part time job |
| 0.34 | hold | 16 | 111 | $S_1$ *hold* down a job |
| 0.34 | knowledge | 16 | 574 | $S_1$: it is common *knowledge* that smoking and passive smoking kill people |

Table 4: Top 12 Word Types Having Wider Meanings in Native (*S*ource) than Non-native (*T*arget) English in ICNALE. $f_S$ and $f_T$: word frequencies in $S$ and $T$, respectively.

be replaced with *almost* as shown in Table 4. Manual investigation reveals that this usage does not appear at all in the non-native portion. This is an example of the robustness of the proposed methods for infrequent instances. This is true for the other example *knowledge*, which appears only 16 times in the native sub-corpus. Out of the 16, according to $r(\boldsymbol{x}, S, T)$, the top two representative word instances of this consist of a *that*-clause describing the formal subject *it* as shown in Table 4. This usage seldom appears in the non-native sub-corpus; as far as we checked manually, only two out of the 574 instances had this construction[9].

### 3.5 Comparison between Chinese and Japanese Learners of English

We now turn our interest to the comparison between non-native speakers of English. Here, we compare the Chinese and Japanese sub-corpora in ICNALE for we understand the two mother tongues at least to some extent.

Table 5 and Table 6 show the top seven word types having wider meanings in the Chinese and Japanese sub-corpora, respectively, which follow the same format as Table 3. It is difficult to examine the results because of the lack of the authors' knowledge about the two languages and the English teaching systems in the respective countries. Keeping in this mind, the results are summarized in the following three points (the details are shown in Appendix C): (i) calque (English words used with a meaning that is transferred from the cor-

responding foreign word and that does not exist in native English), e.g., *meet*; (ii) influence from the sound system of the mother tongue, e.g., *low* for *low* and *law*; and (iii) meaning specialization caused by loan words transliterated from English, e.g., *rest*.

These results together with those in Subsect. 3.4 suggest that the proposed methods might be useful for language learning assistance. An example would be an application to feedback comment generation (Nagata, 2019), which is the task of generating hints or explanatory notes that facilitate language learners. One might be able to find word instances by the detection method and to output pre-defined feedback comments to the results.

### 4 Discussion

The evaluation results in Sect. 3 show that the concentration parameter $\kappa$ or its reciprocal, which is interpreted as a kind of variance, is a good indicator of semantic differences. The proposed detection method, which solely relies on the ratio of $\kappa$, achieves the best detection accuracy as shown in Subsect. 3.2. Besides, the differences shown in Table 3, Table 4, and Table 5 are mostly interpretable and some of them are indeed differences of meaning; after having seen a few extracted representative word instances, we were able to tell, in most cases, where the difference(s) came from.

Interestingly, the results in the SemEval-2020 sub-task suggest that semantic changes are reflected more in the concentration (or variance) than the mean direction $\boldsymbol{\mu}$; the methods that exploit the mean direction do not perform as well as the $\kappa$-based proposed method. It requires more in-

---

[9]We first searched the non-native sub-corpus for the pattern *knowledge that*, obtaining 22 instances. We then looked into them, finding that 19 cases were used in a relative clause and one in an erroneous construction.

| $c(S,T)$ | Word type | $f_S$ | $f_T$ | Representative instance |
|---|---|---|---|---|
| 0.66 | alone | 26 | 52 | $S_1$: *let* alone |
| 0.62 | rest | 23 | 23 | $S_1$: *rest* of cigarettes / $T_1$: take a *rest* |
| 0.60 | gradually | 38 | 27 | $S_1$: $\cdots$ solved *gradually* / $T_1$: $\cdots$ decreasing *gradually* |
| 0.59 | value | 30 | 38 | $S_1$: $\cdots$ *value* the advantages / $T_1$: the *value* of money |
| 0.54 | second | 268 | 330 | $T_1$: *Second,* $\cdots$ |
| 0.53 | contact | 19 | 17 | $S_1$: early *contact* with society / $T_1$: to *contact* with other people |
| 0.49 | meet | 71 | 75 | $S_9$: those restaurants may meet some difficulties |

Table 5: Top Seven Word Types Having Wider Meanings in Chinese ($S$ource) than Japanese ($T$arget) English in ICNALE. $f_S$ and $f_T$: word frequencies in $S$ and $T$, respectively.

| $c(S,T)$ | Word type | $f_S$ | $f_T$ | Representative instance |
|---|---|---|---|---|
| 0.94 | etc | 43 | 14 | $S_5$: clothes and books *etc* |
| 0.73 | contain | 12 | 11 | $S_1$: public place *contain* restaurants |
| 0.68 | word | 18 | 112 | $T_1$: In a *word* |
| 0.62 | taking | 27 | 295 | $T_1$: $\cdots$ taking a part time job can $\cdots$ |
| 0.55 | whose | 13 | 23 | $S_1$: $\cdots$ students whose families are $\cdots$ |
| 0.55 | low | 12 | 15 | $T_3$: the low of prohibitting smoking |
| 0.54 | becoming | 15 | 37 | $S_6$: when becoming a university *student* $\cdots$ |

Table 6: Top Seven Word Types Having Wider Meanings in Japanese ($S$ource) than Chinese ($T$arget) English in ICNALE. $f_S$ and $f_T$: word frequencies in $S$ and $T$, respectively.

vestigations to confirm this argument, but a possible explanation is as follows. Semantic broadening/narrowing normally occurs when a meaning gradually emerges/disappears from the existing meaning(s). Unlike this, the change only in direction (not in variance) would have to assume that a meaning emerges and another disappears to the same degree simultaneously or that a meaning is completely lost and a new one has emerged. Both cases are not so likely in a relatively short period of time as in our 1800s-2000s case.

One of the advantages of the proposed detection method is that it can predict whether the meaning of a target word is narrowing or broadening as illustrated in Subsect. 2.1. Namely, positive and negative values of Eq.(1) indicate narrowing and broadening, respectively. Although it would be difficult to evaluate this property of the proposed method quantitatively, since there is no publicly available dataset in this regard, qualitatively we have seen actual broadening/narrowing word instances in Subsect. 3.3 to Subsect. 3.5.

Another advantage is that the proposed methods are computationally efficient. They require no training or fine-tuning unlike the previous approaches as discussed in Sect. 5. Besides, they have almost no hyper-parameters except for the threshold for word frequency (word instances whose frequency is more than this threshold are the target of analy-sis). They solely rely on an off-the-shelf language model (BERT in our case), which is a large advantage in terms of implementation and development.

Fortunately, $\kappa$, or norms of mean word vectors are stable with respect to word frequency as shown in Appendix D; norms of the mean vector become almost constant with a frequency of five or so. This stability of the norm enables the proposed methods to discover semantic differences in infrequent instances. It should be emphasized that only one word instance would be enough to proof that a word type has a certain meaning, as in the *near* example in Table 4 (while the opposite case does not hold).

It should also be emphasized that its robustness for the low frequency problem comes from the use of contextualized word vectors via a large language model. Even if the source and target corpora are small, the obtained word vectors should be statistically reliable considering the language model is trained on a large corpus. In contrast, the previous methods based on non-contextualized word vectors inevitably suffer from the low frequency problem because non-contextualized word vectors are learned from the input corpora.

As we have discussed, the proposed methods are simple and efficient, but at the same time effective in discovering semantic differences. All these nice properties come from the assumption of the von Mises-Fisher distribution behind word vectors. As

we will discuss in Sect. 7, although this assumption has its limitations theoretically, it works well practically as we have seen in Sect. 3.

## 5 Related Work

Linguists (e.g., Fujimura et al. (2013); McEnery et al. (2019)) often use frequency-based methods to discover differences in words between two corpora. Because they only consider superficial frequency counts, it requires more sophisticated methods to conduct deeper analyses into semantic differences.

The use of non-contextualized word vectors is the major approach to semantic difference detection. For diachronic analysis, Kim et al. (2014) propose setting word vectors obtained from the previous time to initial word vectors of the next. For the same purpose, Kulkarni et al. (2015); Hamilton et al. (2016) propose methods for discovering semantic differences by aligning words in two corpora. These alignment-based methods make a strong assumption that words are linearly aligned between two corpora, which does not necessarily hold in any corpus pair (e.g., native and non-native English). Takamura et al. (2017); Kawasaki et al. (2022) extend this approach to discover semantic differences across languages while their methods require a word-alignment dictionary.

Yao et al. (2018) avoid the problem in word alignment by learning word vectors and alignment simultaneously. Their method has sensitive hyper-parameters that need to be tuned, which results in a complex combinatorial optimization problem (Aida et al., 2021). Dubossarsky et al. (2019) propose a method for detecting semantic differences by simultaneously optimizing multiple word vectors. While this method does not require linear transformations or extensive hyper-parameter search, it requires a list of target words, which is not realistic in practical uses. Aida et al. (2021) extends Dubossarsky et al. (2019)'s method by optimizing multiple context vectors together with multiple word vectors. These non-alignment-based methods, however, still make the assumption that word vectors and/or context vectors are close to each other in two corpora. The task of detecting semantic differences is to find words that are not aligned well in terms of their meanings in two corpora, and thus methods that do not require such assumptions are preferable.

Gonen et al. (2020) propose a method based on nearest neighbors obtained by non-contextualized word vectors to avoid making such assumptions. For this, it is applicable to any pair of corpora. At the same time, it suffers from the bias in corpus sizes and the low frequency problem.

Some researchers use contextualized word vectors for semantic difference detection. Hu et al. (2019); Giulianelli et al. (2020); Rother et al. (2020); Kobayashi et al. (2021) automatically group contextualized word vectors to predict word meanings and then compare the results to detect semantic differences. Predicting word meanings is itself another difficult task. Also, it is costly to train a classifier or to conduct clustering for every single word type found in corpora.

Recently, Aida and Bollegala (2023) have proposed a method exploiting the variance of word vector distribution. Their method models the distribution by the Gaussian distribution and uses both the mean vector and the covariance matrix. The use of the Gaussian distribution has an advantage in that it is not isotropic unlike the von-Mises Fisher distribution. On the other hand, their method is computationally costly. Besides, it requires that words instances should be similar in number in the two corpora for comparison. To avoid the problem, they use sampling to have word instance sets of a similar size, which is not necessary in our methods.

## 6 Conclusions

In this paper, we have proposed using the concentration parameter (or variance), which can be calculated from the norm of mean word vectors, to detect semantic differences with their representative instances. The proposed methods do not require assumptions concerning words and corpora for comparison that the previous methods do. The only assumption is that word vectors follow the von Mises-Fisher distribution. This enables the proposed methods to be applied to corpus pairs such as native and non-native English corpora where the assumptions of the previous methods do not hold. Also, they are simple and efficient in that they do not require training nor extensive hyper-parameter search. With these methods, we have actually discovered semantic differences in historical corpora and also in native and non-native English corpora. We have shown that they are effective even for infrequent word instances and also for corpora whose sizes are considerably different.

## 7 Limitations

It should be emphasized that semantic differences found in two corpora do not necessarily mean that the word type has acquired/lost a meaning (for historical analyses) or the writers do not know/cannot use the missing meaning(s) (for non-native speakers). It would require further investigations to confirm these arguments. Rather, the proposed methods are suitable for obtaining new hypotheses about semantic differences in words or for supporting a hypothesis one already has.

As described in Subsect. 2.3, the proposed methods assume the von-Mises Fisher distribution behind word vectors. This inevitably assumes that the distribution of word vectors is unimodal and isotropic. It can be multimodal and/or anisotropic. A more sophisticated modeling (e.g., a mixture of the von-Mises Fisher distribution (Banerjee et al., 2005)) might achieve further improvements in performance.

The use of the von-Mises Fisher distribution inevitably discards norms of individual word vectors. This does not necessarily mean that norms of word vectors are not important for handling word meanings; they might encode some important aspects of word meanings. Detecting semantic differences is one thing; encoding word meanings in vectors is another. What this paper has shown is that the concentration parameter (or variance) of the von-Mises Fisher distribution is effective in semantic difference detection. More investigations are necessary to reveal what is encoded in vector norms.

Another limitation is that the proposed method assumes that the form of a word is constant as its meaning(s) change. In reality, word forms change as well as meanings. This limitation is common to the previous methods. It is a challenging problem to predict changes in forms and meanings simultaneously.

The use of a large language model to obtain word vectors implicitly assumes that it models the target language well. For such analyses as we conducted in this paper, the assumption should hold at least to some extent. However, it would not hold if the target language differs considerably from the language (data) on which the language model is trained. An example would be ancient languages (or ancient corpora).

## Ethical Statement

We do not foresee any considerable risks associated with the present work. We only use well established datasets for the purposes for which they were designed, which will likely not cause ethical concerns that did not already exist for these data. Nevertheless, it should be emphasized that the methods presented in this paper may return results containing noise. Languages are used differently by different people, and attempts to measure changes in language inevitably simplify the diversity of uses. As such, any work applying the methods to measure semantic change should be aware of their limitations and proceed carefully.

## Acknowledgements

The authors thank all anonymous reviewers and program chairs for their valuable comments. We also thank Hui-Syuan Yeh for her comments on Chinese English. This work was partly supported by JSPS KAKENHI Grant Numbers JP22K12326 and JP23K12152. This work was partly conducted by using computational resource of AI Bridging Cloud Infrastructure (ABCI) provided by National Institute of Advanced Industrial Science and Technology (AIST).

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

## A   Comparing 1800s to 2000s English Corpora

Table 7 shows the list of top ten word types having wider meanings in the 1800s (source) than in the 2000s (target) corpus. It follows the same format of Table 3 (Subsect. 3.3). The word types are classified into the following two categories: transcription errors and potential semantic shift.

**Transcription errors**: They are flagged as a semantic difference because they are seemingly incorrectly transcribed as in the representative word instance of *whore*, which should be *where*; other examples include *teen* for *been*, *coma* for *come*, and *tuna* for *tune*?. Transcription errors increase the variation in the contexts of a word type and shorten its mean word vector. It is crucial to remove transcription errors to conduct accurate analyses. This is especially true for historical corpora. The evaluation results suggest that the proposed detection method may be used to detect transcription errors.

**Potential semantic shift**: All other word types in Table 7 exhibit a potential semantic difference. Although it would be difficult to prove their meaning differences from the available information, they are at least all interpretable. A typical example is *pregnant*. In the 1800s, it means both having a child and rich. In the 2000s, both meanings are still valid, but the ratio has shifted heavily towards the former. From the word instance list sorted by representativeness (i.e., Eq.(3)), we estimate that 99% of the 2,290 instances are of the former sense in the 2000s. The other examples are: *quantum* (a unit vs *quantum* in physics), and *diner* (meal vs an eating place).

Interestingly, while *rebounds* acquires a new meaning in the 2000s, its meanings are narrowing in the new corpus. It refers to something rebounding physically in the 1800s, while it also refers to an action in basketball in the 2000s; according to Wikipedia[10], basketball was first played in 1891 and thus this usage did not exist in the middle 1800s or earlier. This new usage has become the majority, which has made the others relatively less frequent. We cannot tell if this is a true semantic narrowing or for some other reason(s), for example, register differences between the 1800s and 2000s corpora. Still, the above discussion explains why *rebounds* is detected as semantic narrowing in the 2000s although it actually gained a new meaning.

## B   Comparing Non-Native to Native English

Table 8 shows the top 12 word types having wider meanings in the non-native sub-corpora (source) in ICNALE, which follows the same format as Table 4 (Subsect. 3.4). Table 8 reveals the following four major reasons why the word types have wider meanings in the non-native sub-corpus: spelling and grammatical errors; unusual collocations; differences in POS; idiomatic phrases. We describe their details in this order below.

**Spelling and grammatical errors**: Spelling and grammatical errors increase the variety of contexts of a word type. For example, *form* often appears as a spelling error of *from* and thus the superficial word type appears in the contexts of the verb and noun (*form*) and also of the preposition (*from*). This makes the mean vector of the word type shorter. Similarly, *hope* (with a direct object noun) and *responsible* (as a noun or a verb) are used incorrectly.

**Unusual collocation**: Some words are used in unusual collocations (e.g., *I'll introduce interesting job* to mean *recommend*). Similarly, the adjective *great* is often used with negative words such as *harm* and *damage*. This usage is not quite incorrect, but is rare in the native sub-corpus where it is used in a positive sense such as in *a great idea*. Other examples include *section*, and *staff*.

**Differences in POS**: Some words are used with a POS that seldom, or never, appears in the native sub-corpus. For example, *essential* is only used as a predicative adjective as in *It is essential for* in the native sub-corpus, while it is also used as an attributive adjective and a noun in the non-native sub-corpus. Similarly, *pretty* is used as an adverb

---

[10]https://en.wikipedia.org/wiki/Basketball. Accessed on 5th, May, 2023.

| $c(S,T)$ | Word type | $f_S$ | $f_T$ | Representative instance |
|---|---|---|---|---|
| 1.19 | whore | 57 | 483 | $S_1$: the rear, *whore* it was found |
| 1.06 | rebounds | 18 | 342 | $S_2$: the wave *rebounds* / $T_1$: 9.2 *rebounds* |
| 1.01 | teen | 44 | 849 | $S_1$: has already *teen* shown |
| 0.96 | hitter | 40 | 303 | $S_1$: with *hitter* feelings / $T_1$: power *hitter* |
| 0.92 | recession | 32 | 539 | $S_4$: the *recession* of cliffs / $T_1$: After the *recession* began, presidents Bush |
| 0.90 | pregnant | 343 | 2290 | $S_1$: *pregnant* sense / $T_1$ : she was *pregnant*. |
| 0.88 | tuna | 13 | 464 | $S_1$: we *tuna* ourselves with the peoples |
| 0.86 | coma | 30 | 345 | $S_2$: must have *coma* from god. |
| 0.85 | quantum | 43 | 901 | $S_2$: the usual *quantum* of abuse / $T_2$: a *quantum* physicist |
| 0.84 | diner | 14 | 635 | $S_3$: If *diner* was an apple / $T_1$: afternoon shift at the *diner*. |

Table 7: Top 10 Word Types Having Wider Meanings in 1800s (*S*ource) than 2000s (*T*arget) in COHA. $f_S$ and $f_T$: word frequencies in $S$ and $T$, respectively.

| $c(S,T)$ | Word type | $f_S$ | $f_T$ | Representative instance |
|---|---|---|---|---|
| 0.76 | form | 92 | 21 | $S_1$: learn *form* the books / $T_1$: some *form* of |
| 0.72 | degree | 70 | 39 | $S_2$: in some *degree* / $S_3$: to some *degree* |
| 0.64 | pretty | 44 | 28 | $S_5$: *pretty* clothes |
| 0.62 | hope | 195 | 24 | $S_3$: must *hope* money from their parent |
| 0.61 | worry | 94 | 27 | $S_2$: a lot *worry* / $S_5$ their *worry* |
| 0.55 | section | 85 | 11 | $S_3$: important *section* of students / $T_1$: a smoking *section* |
| 0.53 | great | 429 | 67 | $S_1$: *great* harm / $S_2$: *great* damage |
| 0.52 | responsible | 127 | 44 | $S_1$: a man who *responsible* / $S_3$: their *responsible* |
| 0.51 | staff | 45 | 14 | $S_1$: a concert *staff* / $S_4$: a part time *staff* |
| 0.51 | yes | 68 | 13 | $S_1$: If we say *yes*, |
| 0.48 | essential | 86 | 13 | $S_1$: *essential* university students / $S_2$: biochemistry *essential* |
| 0.48 | introduce | 11 | 13 | $S_1$: I'll *introduce* interesting job / $S_2$: *introduce* set |

Table 8: Top 12 Word Types Having Wider Meanings in Non-native (*S*ource) than Native (*T*arget) English (ICNALE). $f_S$ and $f_T$: word frequencies in $S$ and $T$, respectively.

and also as an adjective. It is not clear why the non-native speakers of English use these words with wider POSs, which would be a research question for second language acquisition research.

**Idiomatic phrases**: The non-native speakers often use the idiomatic phrases *to some degree* and *in some degree*, which never appear in the native sub-corpus. In contrast, the native speakers use it to mean a college qualification, which also appears in the non-native sub-corpus. This is similar to *concerned* and *course* in Table 4.

## C Details of Comparison between Chinese and Japanese Speakers of English

The results of the comparison is shown in Table 5 and Table 6 in Subsect. 3.5. Our interpretation of the results are as follows (CHN and JPN refer to the Chinese and Japanese sub-corpora, respectively). Words having wider meanings in CHN:

1. *alone* appears in various contexts including the idiomatic phrase *let alone* in CHN. It ap-

pears only once out of 52 occurrences in JPN. These differences are reflected in their concentration parameters.

2. In CHN, *rest* is used as both *to relax* as in *take a rest* and *remaining things* as in *the rest of the time* while in JPN, it is only used in the first sense. This might be due to the Japanese loan word *resuto* transliterated from *rest*. It only refers to the first sense. Takamura et al. (2017) empirically show that Japanese loan words are used in differently, often in limited meanings, from their English counterparts. This might be the case[11].

3. In CHN, *gradually* collocates with various verbs while in JPN, it does with limited verbs, mostly with *increase*, *decrease*, and *expand* in the progressive form.

4. *value* is used as a noun and a verb in CHN

---

[11]The argument here will be only fully valid after having examined the Chinese case. That is, Chinese may have a loan word such as the Japanese *resuto* and may have limited meanings. This is beyond the authors' ability.

while the verb usage never appears (out of 38 occurrences) in JPN. This might be the influence from the transliterated loan word *baryu* just as in the "rest/resuto" case above.

5. *second* is frequently used in the phrase *Second,* in both CHN and JPN, but the ratio is higher in JPN (68% and 72% in CHN and JPN, respectively). In CHN, it appears in wider varieties of contexts as in *second hand smoke/smoking*.

6. *contact* are used as both a noun and a verb in CHN while it is frequently used as a verb (often with an erroneous preposition (e.g., *contact with*); 12 time out of 17) in JPN.

7. In CHN, *meet* is used to mean *to face* and *to run into* as in *\*meet a difficulty* and *\*meet a problem*. These usages never appear in JPN; rather it is used in the more standard *to see* sense, which also appears in CHN. This might be an influence from the Chinese word 遇到, which is translated into *to face* and *to run into* as well as *to meet*. As a result, erroneous phrases such as the above two appear in CHN.

Words having wider meanings in JPN:

1. *etc* appears only in the sentence end with a comma (e.g., hall, church, etc. EOS) in CHN while it appears everywhere in a sentence in JPN. To be precise, it appears after various noun phrase as in the representative word instance. This is a possible mother tongue interference; the Japanese word *nado* corresponding to "etc" can appear after almost any noun phrase without a comma.

2. Grammatical error, correctly *containing*, which only appears in JPN

3. In CHN, *word* frequently appears in the phrase *in a word* (and the like) (101 times out of 112), which shortens the mean vector and makes its meanings narrower than in JPN. In contrast, it is used in other contexts in JPN.

4. *taking* appears frequently in the phrase *taking a part-time job* in CHN, which is a possible interference from 接受 corresponding *to take* (接受 is used with *job* to mean *to work* or *to do a job* literally in Chinese). The phrase *taking a part-time job*, mostly as a subject, are so frequent that it makes its mean word vector short (and a narrower meanings) in CHN.

5. *whose* appears frequently in the phrase *students whose family are* in CHN, which makes the mean word vector short. The reasons are not clear.

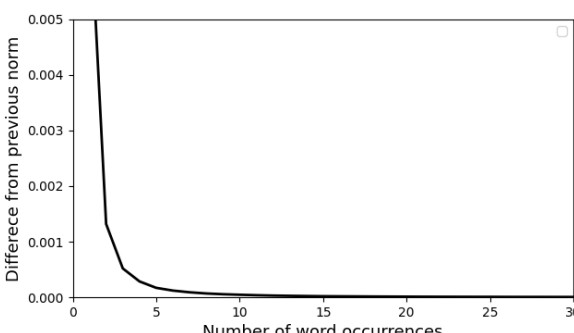

Figure 3: Relationship between Number of Word Occurrences and Differences in Two Consecutive Norms.

6. *low* is used to mean both *low* and *law* in JPN. This is a possible mother tongue interference reflecting the fact that the two sounds pronounced as one sound in Japanese (Swan and Smith, 2001).

7. *becoming* is often used with *more and more* as in the representative instance in CHN. In contrast, it is used with more flexible context including in the progressive with omission as in *when becoming a university student.* in JPN, which never appears in CHNE (out of 37 instances). This might reflect that Chinese learners of English have difficulty with progressive aspect (Swan and Smith, 2001).

## D Relationship between Word Frequency and Vector Norms

We investigated the relationship between word frequency and vector norms. Specifically, we calculated norms of the mean word vector for each occurrence of each word type and then the differences between the two consecutive values of the norms.

Fig. 3 shows the results where the horizontal and vertical axes denote the number of occurrences of word types and the norm differences averaged over all word types, respectively. Fig. 3 shows that after around five occurrences, the average norm difference becomes almost zero, meaning that the norm of the mean vector is almost constant. Considering this, setting the frequency threshold to a small value such as ten gives stable vector norms (and thus stable values of the concentration parameter). This stability of the norm enables the proposed methods to discover semantic differences in infrequent instances.