# OpenReview forum: "Variance Matters: Detecting Semantic Differences without Corpus/Word Alignment"
_EMNLP/2023/Conference — EMNLP 2023 Main_

### Official Review · Reviewer_TMju · 2023-08-04

**Soundness:** 3

**Ethical Concerns:**

Yes

**Excitement:**

3: Ambivalent: It has merits (e.g., it reports state-of-the-art results, the idea is nice), but there are key weaknesses (e.g., it describes incremental work), and it can significantly benefit from another round of revision. However, I won't object to accepting it if my co-reviewers champion it.

**Missing References:**

https://aclanthology.org/P19-1321/
https://paperswithcode.com/paper/learn-interpretable-word-embeddings


**Paper Topic And Main Contributions:**

The paper proposes a method for detecting semantic or usage differences and a method for extracting their typical instances in context based on the variance of contextualised word vectors. The approach should be effective in corpus pairs whose sizes are considerably different and also in infrequent words. The authors propose two methods: one for detecting words that have semantic differences in two corpora and one for extracting the representative instances.

**Reasons To Accept:**

The approach might be useful to compare diachronic corpora and detect periods of creation for some document collections. It could also be applied to detect texts developed by non-native speakers and even discriminate between texts developed by speakers with different mother tongues.

**Reasons To Reject:**

It goes without saying that vectors rely on specific texts, and that some degree of similarity between collections of texts can be measured. The authors stated a few limitations on the suggested approach themselves: the assumption that the form of a word is constant as its meaning(s) change.
Although two-dimensional vectors are specified, it is unclear if this is always the case. Words that are formally identical but have different meanings ought to be connected to various vectors.

**Reproducibility:**

3: Could reproduce the results with some difficulty. The settings of parameters are underspecified or subjectively determined; the training/evaluation data are not widely available.

**Reviewer Confidence:**

3: Pretty sure, but there's a chance I missed something. Although I have a good feel for this area in general, I did not carefully check the paper's details, e.g., the math, experimental design, or novelty.

---

> ### Author Rebuttal · Authors · 2023-08-24
>
> Dear Reviewer TMju,
>
> We would like to express our thanks for the review and the useful information. As pointed out, our explanations are insufficient in some parts of the original manuscript, which makes readers confused. To make it clearer, let us add some more information.
>
> First of all, we thank the reviewer for letting us know about the citations. Both are related to our work and we will cite them in the revised manuscript adequately.
>
> ------
> # Reviewer's comment 1
> > Although two-dimensional vectors are specified, it is unclear if this is always the case.
>
> # Response 1
> (We might be misunderstanding this comment). We suppose that our explanation was confusing. We used the two-dimensional vector case just as an example because we can easily visualize the situations as in Figure 1 and Figure 2 (in the original manuscript). The arguments about the concentration parameter and the length of mean vectors hold for three- or more dimensional vectors. That is, if a word appears in more varieties of contexts (thus, more meanings), its mean word vector gets short. This is shown by Eq.(5) --- concentration parameter $k \approx \frac{l(d-l^2)}{(1-l^2)}$ --- where $l$ and $d$ denote the length of the mean word vector and the dimensionality of the word vector, respectively. Since we did not explicitly state this fact, it was not clear if the arguments hold for higher dimensions. We will add the explanation in the revised manuscript.
>
> # Reviewer's comment 2
> >  Words that are formally identical but have different meanings ought to be connected to various vectors.
>
> # Response 2
> (We might also be misunderstanding this comment). Again we suppose that our explanation seemed to be insufficient on this point, too. As the reviewer suggests, words that are formally identical are mapped to different vectors depending on their surrounding contexts. If their contexts are diverse, the directions of the corresponding vectors are also diverse. Then, summing vectors with diverse directions results in a relatively short summed vector and in turn in a short mean word vector as visualized in Figure 1. We will make this point clear in the revised manuscript.
>
> # Reviewer's comment 3
> > the assumption that the form of a word is constant as its meaning(s) change.
>
> # Response 3
> This is probably the largest limitation in the proposed methods. It is also one of the largest limitations in all previous methods. Considering these facts, we will include the limitation as future work in 6. Conclusions.
>
> ------
>
> Many thanks again for the careful review. We will revise the manuscript following the advice.
>
> Best regards,
>
> Authors.

---

### Official Review · Reviewer_3y58 · 2023-08-05

**Soundness:** 4

**Excitement:**

4: Strong: This paper deepens the understanding of some phenomenon or lowers the barriers to an existing research direction.

**Paper Topic And Main Contributions:**

The authors present a new method for checking whether the same words occurring in two different corpora have the same or different meanings. The method is relatively simple because it uses pre-trained contextual word vectors and is based on counting the norm of the mean vectors. The intuition is that if a word has more meanings (more boundary meanings), the mean vector is shorter. A suitable coefficient for comparing norms in two corpora was defined and called "coverage".  An additional task that was solved was to identify a representative occurrence of a word for a meaning that does not occur in the other corpus.  The solution is based on counting the cosine similarity between a vector representing a particular occurrence of a word and a vector representing the difference between the weighted mean norms. Experiments were conducted on two pairs of corpora: English 1800s/2000s (COHA) and native and non-native speakers of English (ICNALE).  SMEval-2020 Task 1 data (from COHA) was used for quantitative evaluation.  The results obtained were very good, at the level obtained by much more computationally demanding methods.

**Questions For The Authors:**

A: In a sketchy algorithm given in section 2.2 you write that the xs-xt should be counted. But the definition of representativeness  contains weights here.

**Reasons To Accept:**

The proposed method is relatively simple, but very effective. The authors presented the mathematical basis of the proposed measures. An extended qualitative analysis was conducted with some interesting observations about the English language itself.

**Reasons To Reject:**

I do not see any

**Reproducibility:**

4: Could mostly reproduce the results, but there may be some variation because of sample variance or minor variations in their interpretation of the protocol or method.

**Reviewer Confidence:**

3: Pretty sure, but there's a chance I missed something. Although I have a good feel for this area in general, I did not carefully check the paper's details, e.g., the math, experimental design, or novelty.

---

> ### Author Rebuttal · Authors · 2023-08-24
>
> Dear Reviewer 3y58,
>
> We would like to express our thanks for the thorough review. We really appreciate the effort and time.
>
>  As pointed out in the question field,
>
> > In a sketchy algorithm given in section 2.2 you write that the xs-xt should be counted.
> > But the definition of representativeness contains weights here.”
>
> the algorithm in 2.2 was erroneous. It should be:
>
> - Error: For w, compute the difference mean word vector $\bar{x}_S − \bar{x}_T$
> - Correct: For w, compute the mean word vectors $\bar{x}_S$ and $\bar{x}_T$
>
> and the rest are unchanged. We will correct the error in the revised manuscript.
>
> Many thanks again for the careful review.
>
> Best regards,
>
> Authors

---

### Official Review · Reviewer_Nr3G · 2023-08-10

**Soundness:** 4

**Excitement:**

4: Strong: This paper deepens the understanding of some phenomenon or lowers the barriers to an existing research direction.

**Paper Topic And Main Contributions:**

This paper proposes methods for measuring semantic differences in words between two corpora, contributing to computationally-aided linguistic analysis.

Using a pre-trained large language model, e.g., BERT, new methods are used to examine the coverage of the meanings of words, through the norm of the mean word vectors across meanings. The proposed methods are easy to implement and without pre-training any language models, and do not require alignments between words or corpora for comparison. The proposed methods excel in semantic difference detection tasks compared to previous works, reveal potential semantic shift and difference in POS using diachronic datasets. Furthermore, the differences between native and non-native English usages are revealed using the proposed methods, in terms of different aspects, which can be beneficial for further investigation in second language acquisition.

The paper also discusses the limitations of the proposed methods, specifically the presumption of using von Mises-Fisher distribution and also other limitations for applications.

**Reasons To Accept:**

1. The proposed methods present a straightforward and effective approach for detecting semantic differences and the mathematical background is illustrated in certain detail.
2. Use-cases are given with both quantitative and qualitative analyses.
3. The codes are provided and the work is reproducible.
4. The work can be beneficial in research in other disciplines, such as second language acquisition.

**Reasons To Reject:**

1. This work has only done experiments and reported results in English corpora, it would be very interesting to see some analysis in different languages or across languages, especially in the non-native English users with their mother tongue languages, and explore further potential in using variance metrics in second language acquisition research.
2. As mentioned in the limitation section, that the usage of a large language model to obtain word vectors implicitly assumes that it models the target language as well. The methods rely on a monolingual language model trained on certain time period of data can be limited in detecting semantic shifts across a wider period of time. The current approach can be improved and this topic can be explored further.

**Reproducibility:**

4: Could mostly reproduce the results, but there may be some variation because of sample variance or minor variations in their interpretation of the protocol or method.

**Reviewer Confidence:**

3: Pretty sure, but there's a chance I missed something. Although I have a good feel for this area in general, I did not carefully check the paper's details, e.g., the math, experimental design, or novelty.

---

> ### Author Rebuttal · Authors · 2023-08-24
>
> Dear Reviewer Nr3G,
>
> We would like to express our thanks for the review and the useful comments. We would be grateful if you could read and check the following responses.
>
> -------
> # Reviewer’s Comment 1:
> >This work has only done experiments and reported results in English corpora, it would be very interesting to see some analysis in
> > different languages or across languages, especially in the non-native English users with their mother tongue languages, and explore
> > further potential in using variance metrics in second language acquisition research.”
>
>
> # Response 1:
> We appreciate the suggestion. Following the advice, we have done comparisons between different non-native English users. As a result, we were able to find meaning/usage differences; some of them are possibly ascribed to mother tongue interference. The details are shown below, but they are summarized in three points: (i) influence from the sound system of the mother tongue, (ii) meaning specialization caused by words transliterated from English, (iii) calque (English words used with a meaning that is transferred from the corresponding foreign word and that does not exist in native English). It is difficult to attest them because of the lack of our knowledge about the languages and the English teaching systems in the respective countries. However, the results suggest that the proposed methods are likely beneficial for further investigations in second language acquisition as the reviewer suggests.
>
>
> The details of the additional experiment are as follows.
> - Data:
>  ICNALE six non-native sub-corpora: Chinese, Indonesia, Japanese, Korea, Thai, Taiwanese English data abbreviated as CHN, IND, JPN, KOR THA, TWN, respectively, hereafter.
>
> - Method:
> We have mainly compared CHN and JPN for we understand the two mother tongues to some extent. Specifically, we have extracted words having wider meanings/usages for the combination and have extracted their representative word instances just as in the experiments we described in the original manuscript. We have investigated top seven words for CHN and JPN. We compared them with other sub-corpora in some cases.
>
> - Results: The two tables below with their explanations.
>
> Table: Words having wider meanings in JPN (w.r.t CHN)
> -------
> |TopN| Word type |  Representative instance|
> | ---- | ---- | ---- |
> |1| etc| study, club activity , part time jobs _etc_ all things are a part of college life.|
> |2.|	contain| public place _contain_ restaurants
> |3|word| in a _word_ (in CHN)
> |4|taking| _taking_ a part-time job (in CHN)
> |5|whose| students _whose_ family are (in CHN)
> |6|low| the _low_ of prohibiting smoking
> |7|becoming| when _becoming_ a university student
>
>
> Corresponding interpretation and explanation
> 1. “etc” appears only in the sentence end with a comma (e.g., hall, church, etc. EOS) in CHN (similarly in IDN and TWH) while it appears everywhere in a sentence in JPN. To be precise, it appears after various noun phrase as in the representative word instance above. This is a possible mother tongue interference; the Japanese word “nado” corresponding to “etc” can appear after almost any noun phrase without a comma.
> 2. Grammatical error, correctly “containing”, which only appears in JPN
> 3. In CHN, “word” frequently appears in the phrase “in a word” (and the like) (101 times out of 112), which shortens the mean vector and makes its meanings narrower than in JPN. In contrast, it is used in other contexts in JPN.
> 4. “taking” appears frequently in the phrase “taking a part-time job” in CHN, which is a possible interference from 接受 corresponding “to take” (接受 is used with “job” to mean “to work” or “to do a job” literally in Chinese). The phrase “taking a part-time job” so frequently  that it makes its mean word vector short (and a narrower meanings in CHN).
> 5. “whose” appears frequently in the phrase “students whose family are” in CHN
> 6. “low” is used to mean both “low” and “law” in JPN. This is a possible mother tongue interference reflecting the fact that the two sounds pronounced as one sound in Japanese (Swan and Smith, Learner English, Cambridge University Press, 2001). Similarly, the same error appears in KOR, which is detected compared to CHN by the proposed methods. This agrees the fact that Korean learners of English a similar problem in English vowels (Ibid.)
> 7. “becoming” is used in the progressive form in JPN as in “when becoming a university student.”
>
>
> Table: Words having wider meanings in CHN (w.r.t JPN)
> -------
> |TopN| Word type |  Representative instance|
> | ---- | ---- | ---- |
> |1|alone| let _alone_
> |2|rest| take a _rest_ (in JPN)
> |3|gradually| is decreasing _gradually_ (in JPN)
> |4|value| Companies _value_ work experience
> |5|second| _Second_, (in JPN)
> |6|contact| early _contact_ with society
> |7|meet| _meet_ a similar problem
>
> Corresponding interpretation and explanation
> 1. “alone” appears in various contexts including the idiomatic phrase “let alone” in CHN. The distribution is similar in JPN, but the idiomatic phrase appears only once out of 52 occurrences. These differences are reflected in the concentration parameters.
> 2. “rest” is used in two meanings (“to relax” as in “take a rest” and “remaining things” as in “the rest of the time”) in CHN. In contrast, it is only used to mean “to relax” in JPN. This might be a possible mother tongue interference; influence from the transliterated Japanese “resuto” that only refers to the first meaning. Because of this, most of the Japanese learners in ICNALE might have only known the word “rest” with the first meaning and have used it frequently.
> 3. In CHN, “gradually” collocates with various verbs while in JPN, it  is used frequently with “increase”, “decrease”, and “expand” in the progressive form as in the above example.
> 4. “value” is used as a noun and a verb in CHN while the verb usage never appears (out of 38 occurrences) in JPN. This might be a possible mother tongue interference just as in the “rest/resuto” transliteration influence above; “value” is transliterated as “baryu” in Japanese, which is only used as a noun.
> 5. “second” is frequently used in the phrase “Second, “ in both CHN and JPN, but the ratio is higher in JPN (68% and 72% in CHN and JPN, respectively). In CHN, it appears in wider varieties of contexts as in “second hand smoke/smoking”.
> 6. “contact” used as both a noun and a verb in CHN while it is frequently used as a verb (often with an erroneous preposition (i.e., contact with); 12 time out of 17) in JPN.
> 7. In CHN, “meet” is used as “face” and “run into” as in “face a difficulty” and “run into a problem”. These usages never appear in JPN; rather it is used in the more standard “to see” sense, which also appears in CHN. This might be a influence from the Chinese word 遇倒, which is translated into “to meet”, “to face”, and “to run into”.
>
> # Reviwer’s Comment 2
> > As mentioned in the limitation section, that the usage of a large language model to obtain word vectors implicitly assumes that it
> > models the target language as well. The methods rely on a monolingual language model trained on certain time period of data can be
> > limited in detecting semantic shifts across a wider period of time. The current approach can be improved and this topic can be explored
> > further.:
>
> # Response 2
> As the reviewer points out, the limitation concerning language modeling is one of the challenges we have to attack in the future work. Luckily, the first comment and this second comment gave us a chance to rethink the limitation. Now we feel that the limitation has been somewhat relaxed thanks to the comments.
>
> The current language model, BERT, is trained on English Wikipedia and BookCorpus, both of which should consist of rather recent texts. Wikipedia might contain webpages citing texts published in the 1800s or earlier, but the majority should be those written more recently. Although the details of BookCorpus are not available, the corresponding paper (Zhu et al., 2015) says that it consists of unpublished books, which suggests that they were written again recently. To sum up, BERT is mainly trained on recent texts and a relatively small amount from the 1800s or earlier. Nevertheless, the experimental results in 3.2 and 3.3 (in the original manuscript) show that the BERT-based proposed methods are effective in detecting words with semantic differences in the 1800s. This supports the hypothesis that language models can work well for texts in different time periods as long as the time difference is within a certain amount such as 100-200 years as in the present case.
>
> A similar argument can be made about the comparison between native and non-native English data. Again, BERT is mainly trained on native English data and not on non-native one; of course, Wikipedia (and probably BookCorpus, too) should contain texts written by non-native speakers of English, but the vast majority are of native speakers. Still, the BERT-based proposed methods are effective in native and non-native English comparison. Besides, we have just seen that they are also effective in the non-native-non-native comparisons (between non-native speakers of English with different mother tongues) as shown in the first response. These results support the argument that language models work well (with the proposed methods) on texts of the target language as long as the difference between the training and target corpora is not too large.
>
> From these, we now speculate that language models do not even have to model the target sub-language well in the sense that they predict masked words or following words when they are used as a part of the proposed methods. All they should have to do is to map word instances to vectors of different directions. Even when they are given an anormal context such as an ancient sentence or an unnatural, erroneous sentence, they only have to map it to a vector with a direction different from the others, which is not so hard to believe.
>
> Having said that, this is only our speculation. We need more investigations to confirm this argument. We do not know how much difference in time period or language uses language models can afford when they are used with the proposed methods. These should be included in our future work.
>
> We will include the above discussion in the conclusion section and include the new challenges in our future work. The reviewer’s comments and the discussion concerning them have deepened our understanding of the proposed methods and in turn have enriched the contents of the manuscript. We really appreciate your thoughtful suggestions and expertise.
>
>
> ------
>
> Many thanks again for the careful review. We will revise the manuscript following the advice.
>
> Best regards,
>
> Authors.

---

### Meta-Review · Area_Chair_mHaF · 2023-09-19

**Recommendation:** 5

**Metareview:**

The paper presents simple yet efficient methods that can be used to compare word usage across different contexts (historical, native vs. non-native etc.). The methodology is sound and uncomplicated and the results are compelling.

Work that identifies simple techniques that allow us to capture significant linguistic properties and better understand the structure of human language will make a useful contribution to this track.

---

### Decision · Program_Chairs · 2023-10-07

**Decision:**

Accept-Main

**Comment:**

The paper presents simple yet efficient methods that can be used to compare word usage across different contexts (historical, native vs. non-native etc.). The methodology is sound and uncomplicated and the results are compelling.

Work that identifies simple techniques that allow us to capture significant linguistic properties and better understand the structure of human language will make a useful contribution to this track.